# Molecularly targeted photothermal ablation improves tumor specificity and immune modulation in a rat model of hepatocellular carcinoma

Nina M. Muñoz[1], Crystal Dupuis[1], Malea Williams[1], Katherine Dixon[1], Amanda McWatters[1], Rony Avritscher[1], Richard Bouchard[2], Ahmed Kaseb[3], Kyle M. Schachtschneider[4,5,6], Arvind Rao[7,8,9,10] & Rahul A. Sheth [ID] [1✉]

Thermal ablation is a standard therapy for patients with hepatocellular carcinoma (HCC). Contemporary ablation devices are imperfect, as they lack tumor specificity. An ideal ablation modality would generate thermal energy only within tumoral tissue. Furthermore, as hyperthermia is known to influence tumor immunity, such a tumor-specific ablation modality may have the ability to favorably modulate the tumor immune landscape. Here we show a clinically relevant thermal ablation modality that generates tumor-specific hyperthermia, termed molecularly targeted photothermal ablation (MTPA), that is based upon the excellent localization of indocyanine green to HCC. In a syngeneic rat model, we demonstrate the tumor-specific hyperthermia generated by MTPA. We also show through spatial and transcriptomic profiling techniques that MTPA favorably modulates the intratumoral myeloid population towards tumor immunogenicity and diminishes the systemic release of oncogenic cytokines relative to conventional ablation modalities.

[1] Department of Interventional Radiology, The University of Texas MD Anderson Cancer Center, Houston, TX, USA. [2] Department of Imaging Physics, The University of Texas MD Anderson Cancer Center, Houston, TX, USA. [3] Department of Gastrointestinal Medical Oncology, The University of Texas MD Anderson Cancer Center, Houston, TX, USA. [4] Department of Radiology, University of Illinois at Chicago, Chicago, IL, USA. [5] Department of Biochemistry & Molecular Genetics, University of Illinois at Chicago, Chicago, IL, USA. [6] National Center for Supercomputing Applications, University of Illinois at Urbana-Champaign, Urbana, IL, USA. [7] Department of Computational Medicine & Bioinformatics, University of Michigan, Ann Arbor, MI, USA. [8] Department of Radiation Onology, University of Michigan, Ann Arbor, MI, USA. [9] Department of Biostatistics, School of Public Health, University of Michigan, Ann Arbor, MI, USA. [10] Department of Biomedical Engineering, University of Michigan, Ann Arbor, MI, USA. ✉email: RASheth@mdanderson.org

While mortality from most cancers has decreased in the past two decades, both the incidence and mortality from hepatocellular carcinoma (HCC) are increasing rapidly[1]. Intrinsic tumor resistance and underlying liver dysfunction limit the efficacy and safety of systemic chemotherapies[2]. As a result, there is a dearth of systemic treatment options for patients with HCC[3]. For patients with small HCC lesions, though, local tumor control with thermal ablation such as radiofrequency ablation (RFA) is highly effective and is the standard of care[4].

Contemporary ablation modalities, including RFA, however, are imperfect for several reasons. Although these procedures are typically performed under imaging guidance using computed tomography (CT) imaging, target lesions may be poorly visualized; moreover, the ablation devices have no methods for confirming appropriate positioning within the target lesion[5]. Furthermore, all current ablation modalities lack specificity for ablating tumor versus normal tissue. Thermal energy is delivered indiscriminately around the ablation needle, regardless of whether the needle is positioned within the lesion or not. As a result, tumors may be incompletely treated, and/or a substantial amount of adjacent liver parenchyma may be thermally injured.

One immediate ramification of off-target ablation is damage to adjacent tissue, including potentially critical structures such as bile ducts, portal vein, diaphragm, or peritoneum. In addition, a less immediately apparent but important sequela of off-target ablation is systemic tumor stimulation. RFA has been shown to cause distant tumor growth following hepatic ablation procedures in preclinical primary and metastatic liver tumor models[6–10]. A key factor for these adverse effects is the response of the liver parenchyma that is included in the ablation zone. Upon exposure to hyperthermia, hepatocytes release multiple signaling molecules including extracellular heat shock proteins[8,11–14] and upregulate a host of inflammatory cytokines including tumor necrosis factor-α (TNFα) and interleukin (IL)-6[15,16] that can drive inflammation-mediated carcinogenesis and result in local and remote tumor growth[6,17].

An ideal ablation modality would have the ability to provide the interventionalist with real-time feedback regarding the accuracy of the ablation needle's positioning within the tumor. It would also exhibit tumor specificity, generating thermal energy only within tumoral tissue.

Such a modality has the potential to overcome the oncogenic side effects of currently available technologies. By generating hyperthermia within the tumor tissue but not in the adjacent liver parenchyma, the elaboration of deleterious growth factors would be diminished relative to conventional modalities. Furthermore, a tumor-specific ablation modality may have the ability to favorably modulate the tumor immune landscape. Although ablation traditionally has had minimal relevance for patients with advanced HCC, there are burgeoning clinical and preclinical data indicating that hyperthermia can overcome immunosuppressive barriers and thus improve clinical response rates to systemic immunotherapies for this patient population[18–20].

To address this unmet need, we have developed a clinically relevant thermal ablation modality that generates tumor-specific hyperthermia. Termed molecularly targeted photothermal ablation (MTPA), this technology builds upon prior work in which we have demonstrated that the clinically available near-infrared (NIR) fluorescent drug indocyanine green (ICG) to HCC with exceptional target-to-background ratios (TBRs) (median 7:1)[21,22]. Like most organic fluorochromes, ICG converts the majority of incident light (~90%[23]) to heat. ICG is also thermally unstable[24] and degrades at cytotoxic temperatures. Thus, ICG provides (1) imaging guidance by illuminating tumoral tissue; (2) tumor-specific heat generation; and (3) auto-regulation of the ablation zone by degrading once cytotoxic temperatures are reached. The

purpose of this study was to evaluate MTPA's ability to achieve local tumor control in phantom and in vivo studies. We also evaluated the putative advantages of tumor-specific hyperthermia on the tumor microenvironment relative to RFA through serologic, cell-based, and transcriptomic assays.

## Results

**In vitro and in vivo evaluation of MTPA as a thermally tunable, tumor-specific hyperthermia modality.** We first performed in vitro experiments to determine whether the heat generated by illuminating ICG with NIR light was sufficient to cause clinically relevant temperature increases at laser powers that spare normal tissue. In a gelatin phantom and ex vivo bovine liver tissue, ICG-specific temperature generation was achievable even at concentrations as low as 0.05 mg/mL. Lethal temperatures were readily achievable with higher concentrations: 0.4 mg/mL at 2 W generated temperature increases of 66.2 °C and 56.7 °C in ex vivo liver and gelatin phantom, respectively. As expected, the degree of hyperthermia varied as a function of laser power and ICG concentration. There was minimal (0.2–0.3 °C) temperature increase within the adjacent non-ICG containing phantom (Fig. 1).

Next, we evaluated the in vivo tumor specificity of MTPA in an orthotopic rat HCC model following the intravenous administration of ICG (n = 5 per group). As previously seen in human studies, there were improved TBRs at longer timepoints owing to wash-out of ICG from normal liver parenchyma but sustained retention of ICG within the tumor. TBRs at 24, 48, and 96 hours post ICG injection were 0.5 ± 0.1, 0.68 ± 0.1, and 3.78 ± 0.3, respectively. Based on these data, we then performed MTPA by illuminating liver tumors and the adjacent normal liver parenchyma with 785 nm NIR light at 96 hours post ICG injection (n = 10). Hyperthermia developed within the tumors, whereas significantly less heat generation occurred within normal liver (P < 0.001) (Fig. 1). Tumoral hyperthermia was again found to be tunable by adjusting the laser power (0–400 mW), with temperatures ranging from 37 °C to >100 °C. When MTPA was performed at laser powers to generate lethal hyperthermia (>60 °C), post-procedure imaging, as well as histology, confirmed successful local tumor control akin to conventional thermal ablation modalities (Fig. 1).

**MTPA and RFA differentially alter the local tumor immune microenvironment.** Rats with orthotopic tumors (n = 5 per arm) were randomized to undergo MTPA, RFA, or sham surgery. Tumors were then harvested on postoperative day 2, and immune profiling was performed using immunohistochemistry, flow cytometry, digital spatial profiling (DSP), bulk RNA sequencing (RNAseq), and single-cell RNA sequencing (scRNAseq). Flow cytometry (Fig. S1) demonstrated that both MTPA and RFA increased the local CD3+ T-cell population within the tumor relative to sham, and that the plurality of this population was comprised of CD3+ CD8+ effector T cells (Fig. 2) (% CD45+ /CD3+/CD8+ T cells: RFA 37.3%, MTPA 48.5%, sham 1.1%).

For a more comprehensive multiplex analysis of the immune profile across discrete spatial landmarks, we performed DSP of tumors treated with RFA and MTPA (Fig. 3). Within the tumor tissue, there were significant alterations to the tumor microenvironment compared with baseline for both hyperthermia modalities as well as between the two hyperthermia modalities themselves. Both modalities reduced tumor growth as reflected in decreased Ki67. Both modalities also increased the intratumoral localization of CD11b+ and CD163+ myeloid cells, though the increase of the latter was more significant following MTPA compared with RFA. Interestingly, MTPA resulted in a decrease in β-catenin expression within the tumor, whereas RF ablation

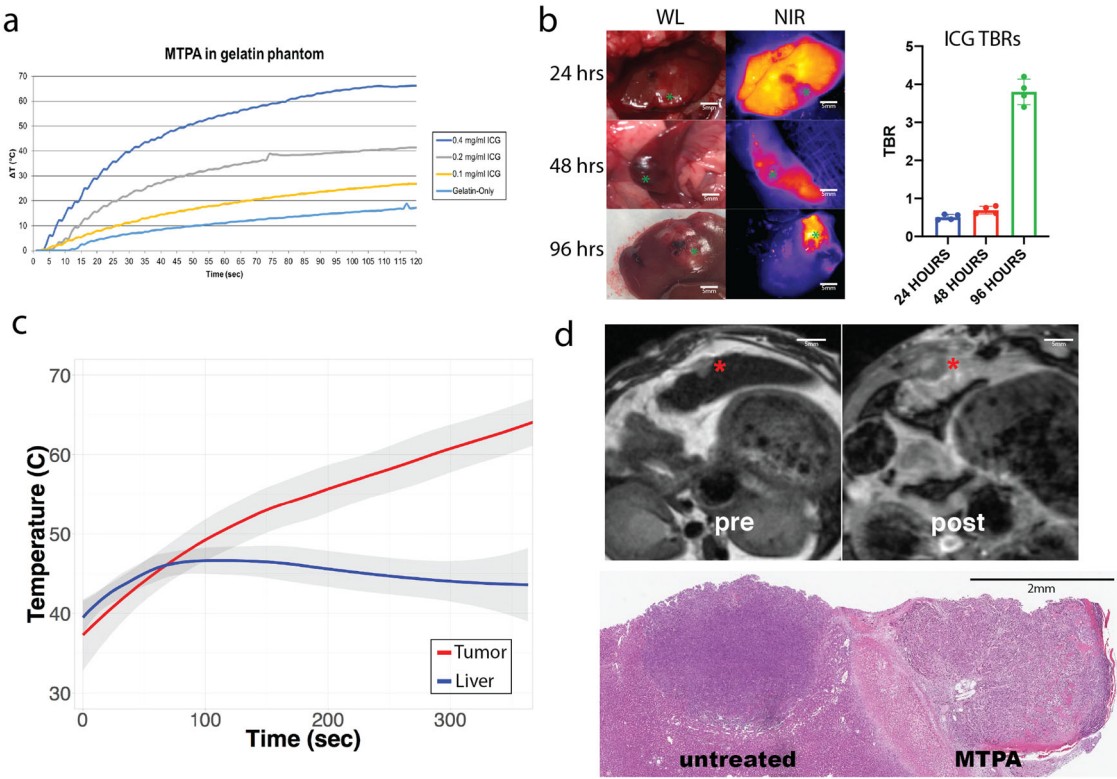

**Fig. 1 In vitro and in vivo evaluation of MTPA. a** In a gelatin tissue phantom, various concentrations of ICG were illuminated with an 808 nm NIR laser at 2 W. A thermocouple adjacent to the ICG was used to acquire continuous temperature data. ICG-specific temperature generation upon illumination by NIR light was achievable even at low ICG concentrations. **b** Time course study of ICG localization in an orthotopic, syngeneic rat model of HCC revealed improved TBRs at late timepoints, with a TBR of ~4:1 when imaged at 96 hours post i.v. injection. $N = 4$ animals per group. **c** In vivo illumination of orthotopic HCC tumors with a 785 nm laser at 450 mW in animals injected with i.v. ICG 96 hours prior generated ablative temperatures within the tumor but not in the adjacent liver parenchyma, thus highlighting the tumor specificity of MTPA. $N = 4$ animals. **d** MRI images prior to ablation (pre, T2-weighted image) demonstrates a small focal HCC. Following MTPA, a zone of ablation completely surrounds the lesion (post, T1-weighted post-contrast fat-saturated image). Histologic examination following MTPA of an animal with two focal orthotopic HCC tumors, one of which was targeted with MTPA, shows a zone of ablation encompassing the entire treated lesion.

did not. Both modalities increased expression levels for several known immune checkpoints such as PD-L1 and VISTA. At the liver tumor interface, both RFA and MTPA decreased immunosuppressive markers such as FoxP3, indicating a decrease in T regulatory cells following either ablation modality. However, there were significant distinctions in multiple immunologic markers between the two modalities. For example, there were increased CD11b+ myeloid cells, fewer "classical" CD14+ monocytes, fewer CD127+ "naive" T cells, and fewer Lag3+ and Tim3+ "exhausted" T cells with MTPA relative to RFA.

**MTPA modulates the intratumoral myeloid compartment towards immunogenic phenotypes**. To further investigate the transcriptional changes to the tumor immune microenvironment, we performed scRNAseq on the intratumoral CD45+ cell myeloid population following MTPA, RFA, and sham surgeries of orthotopic HCC tumors (Fig. 4). This revealed a significant shift away from the M2 phenotype towards the M1 phenotype in myeloid cells isolated from orthotopic tumors treated with MTPA compared with RFA.

As interferons are potent drivers of M1-associated gene transcription, and as activation of a type I interferon response downregulates β-catenin in HCC cells[25], we hypothesized that MTPA effects immunogenic modulation through type I interferon signaling. To test this, we performed bulk RNA sequencing on tumor tissue treated with MTPA versus sham surgery. After identifying differentially expressed genes, we focused on 487

known interferon-stimulated genes (ISGs) that are induced by activation of a type I interferon response (Fig. 4). This revealed significant upregulation of numerous ISGs, indicative of a robust type I interferon response following MTPA compared with sham.

To further investigate the influence of hyperthermia on the interferon-STAT pathway and β-catenin, we performed in vitro analysis of sublethal hyperthermia on mouse and rat HCC cell lines (Fig. 5). Hyperthermia to 43 °C resulted in a decrease in β-catenin expression relative to control; this effect was abrogated by the use of the STAT3 inhibitor Stattic.

**MTPA ameliorates the local and systemic oncogenic effects of RFA**. As previously shown by others, RFA of liver tumors activates both local and systemic oncogenic factors[6,9]. To characterize the differences in gene activation following RFA and MTPA within the treated tumor itself, we performed bulk RNA sequencing of orthotopic tumors treated with both modalities. A total of 209 differentially genes were identified between tumors treated with RFA versus MTPA. Interestingly, this list of genes included several canonical growth factor pathways for HCC; for example, there were significant fold increases in the expression of genes associated with the epidermal growth factor pathway (2.8 log fold change, adjusted $P = 0.02$), insulin-like growth factor pathway (3.2 log fold change, adjusted $P = 0.03$), and the c-Met/ hepatocyte growth factor pathway (1.9 log fold change, adjusted $P = 0.03$) following RFA compared with MTPA. The association

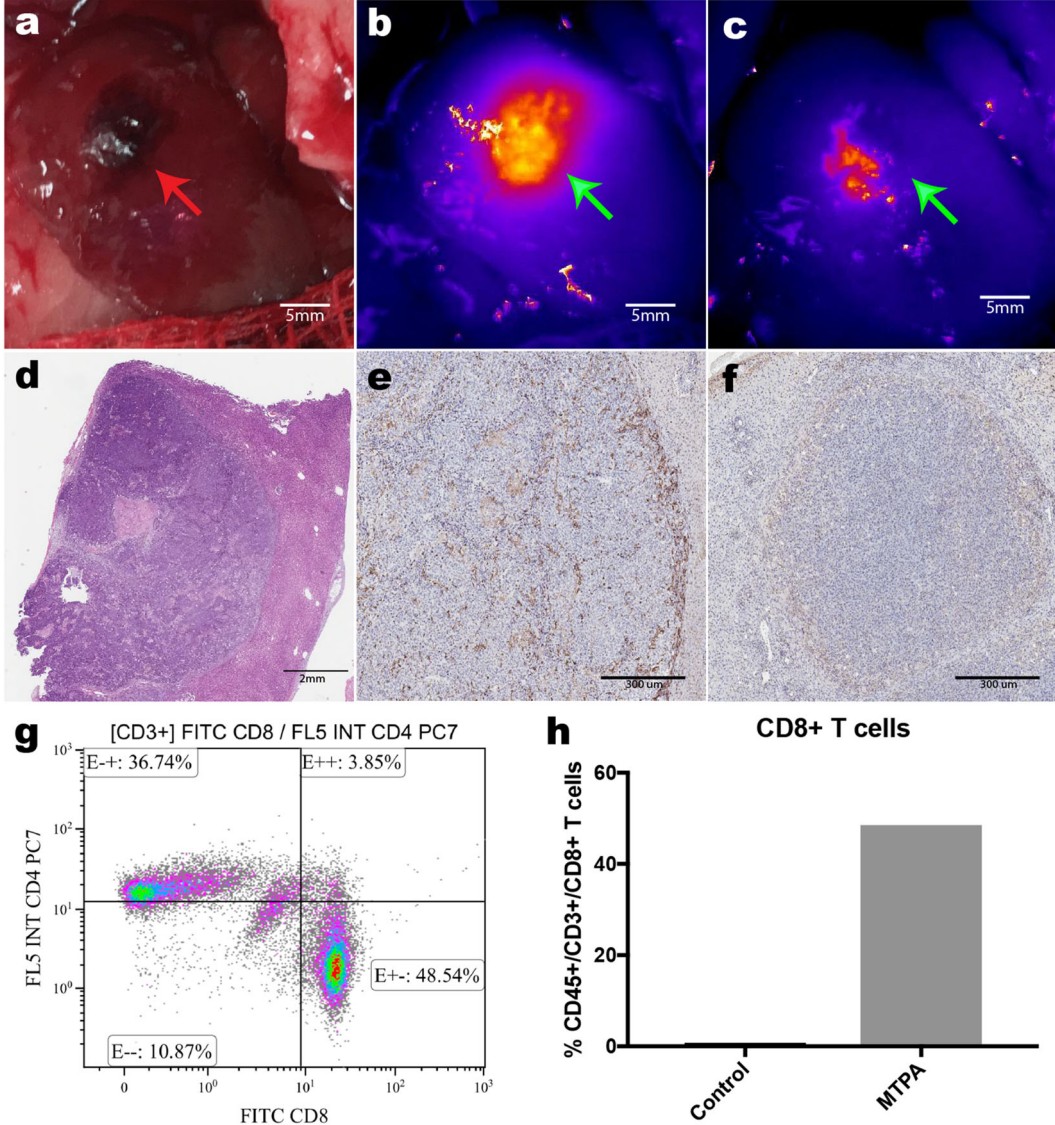

**Fig. 2 MTPA increases intratumoral T-cell infiltration. a** After minilaparotomy, the orthotopic HCC (red arrow) was elaborated to the skin surface. **b**, **c** NIR fluorescence prior to (**b**) and following (**c**) MTPA showed focal ablation of the right side of the tumor as a loss of ICG fluorescence (green arrow). **d** H&E staining shows patchy areas of necrosis (pink). E, CD8+ infiltration (brown cells) is increased following MTPA compared with a control tumor on immunohistochemistry (**f**). Flow cytometry (**g**, **h**) confirms the significant increase in CD3+ CD8+ cell infiltration with MTPA.

of these pathways is in concordance with several recent studies[6,26].

Furthermore, to better understand the potential oncogenic effects of thermal injury to normal liver parenchyma, we performed RF ablation of normal hepatic parenchyma and analyzed the transcriptional changes with RNA sequencing. This revealed 84 differentially expressed genes. Functional analysis of these genes using the DAVID toolkit[27,28] identified multiple biological processes associated with wound healing, innate immunity, and oncogenesis through cytokines such as TNFα (Table 1).

Next, we evaluated the influence of MTPA and RFA on systemic oncogenesis (Fig. 6). Rats with simultaneously implanted orthotopic and subcutaneous tumors were treated with MTPA, RFA, or sham surgery of the orthotopic tumor. Multiplex cytokine analysis from serially acquired serum samples revealed that animals treated with RFA had a significant increase in several cytokines associated with systemic oncogenesis including IL-1α, IL-6, and TNFα; animals treated with MTPA had no significant increase of these cytokines relative to sham. The influence of this cytokine release manifested as accelerated growth of the subcutaneous tumors in animals treated with RFA relative to control, whereas the subcutaneous tumors in MTPA-treated animals grew at the same rate as prior to the intervention and similar to control animal tumors.

## Discussion

We have developed a clinically translatable photothermal therapy modality termed MTPA that exhibits robust tumor specificity for HCC as well as favorable local and systemic immunologic ramifications relative to the current standard of care RFA. Although conceptually related ablation modalities such as laser-induced interstitial therapy[29,30] and photodynamic therapy exist, these approaches have minimal to no tumor specificity. The fact that MTPA relies on clinically approved drugs and devices reinforces its clinical translatability and underlines the technology's potential for clinical impact.

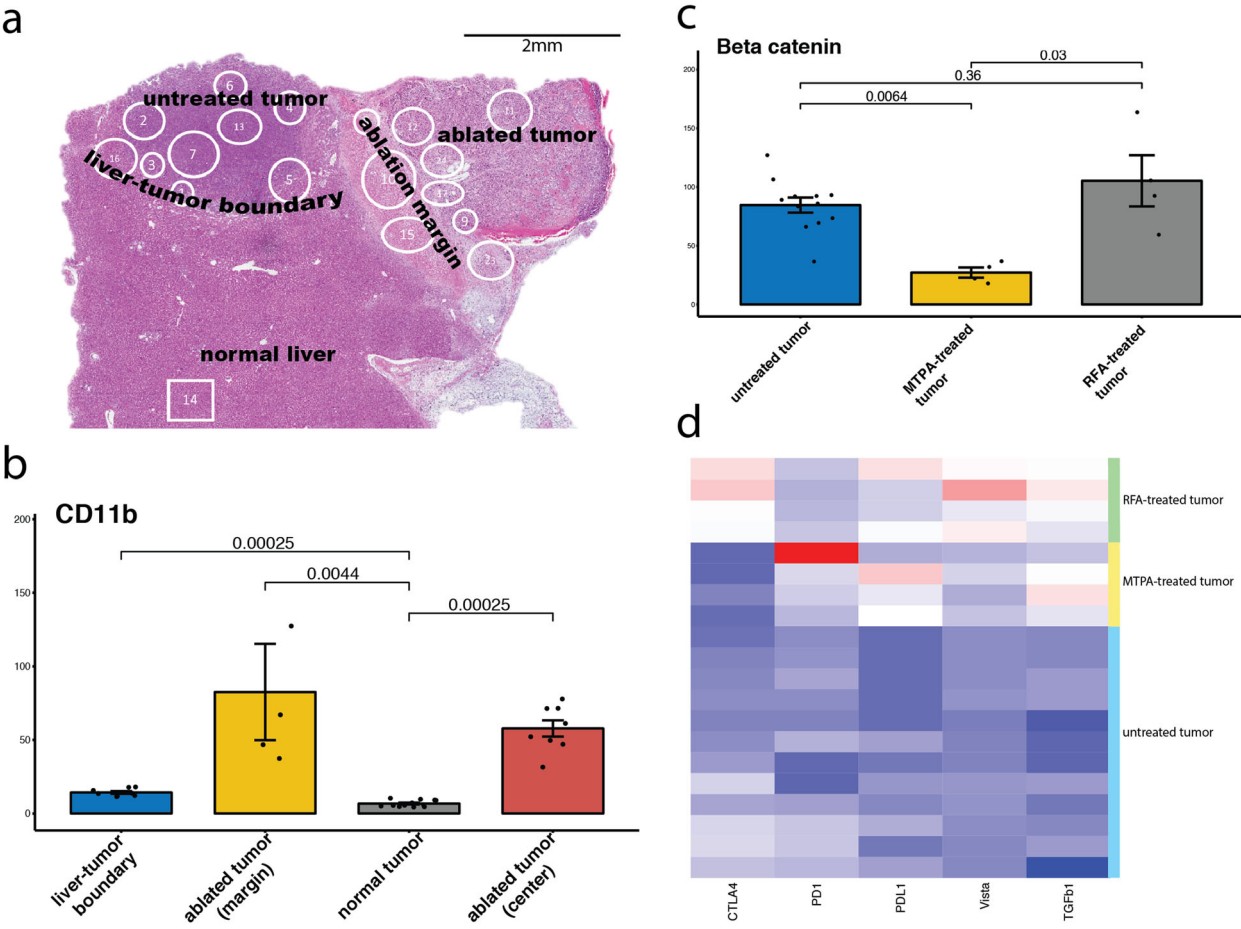

**Fig. 3 Quantitative multiplex spatial immunoprofiling.** Nanostring DSP was performed to characterize >40 immunologically relevant proteins across multiple regions of interest (ROIs) in orthotopic HCC tumors treated with MTPA or RF hyperthermia. Each manually drawn circle or square represents a discrete ROI (**a**). There was a significant increase in myeloid cell markers such as CD11b (**b**) within the margin as well as the center of the ablated tumor following MTPA (**b**). Compared with RFA and untreated tumor, MTPA resulted in a decrease in beta-catenin expression (**c**). However, both ablation modalities resulted in an increased immune checkpoint expression (**d**). $N = 20$ independent measurements.

HCC is the third most common cause of cancer-related deaths globally, and in the United States, it is the fastest-growing cause of cancer-related deaths[31]. For patients with early-stage disease, local control through thermal ablation modalities such as RFA is the standard of care. Although RFA outcomes approach surgical resection outcomes for local disease control of small HCC lesions[4], there is now a growing awareness of the profound local and systemic inflammatory response than follows these interventions. Conventional RFA technique dictates that, in order to maximize local control, ablation zones should include not just the tumor but also at least 5–10 mm of adjacent normal parenchyma[32]. However, the intentional injury to the normal liver for the ablation margin, coupled with RFA's lack of tumor specificity, elaborates a wound-healing response that can drive oncogenesis[9]. Indeed, RFA has been shown to cause distant tumor growth following hepatic ablation procedures in numerous preclinical primary and metastatic liver tumor models[6–10]. This effect is driven by several oncogenic pathways, including the hepatocyte growth factor/c-Met pathway[9,17,33]. This pathway's activation results in the release of inflammatory cytokines including IL-6[15,16] that can drive inflammation-mediated carcinogenesis and result in local and remote tumor growth[6,17]. In corroboration with these studies, we identified upregulation of growth factor pathways following thermal injury with RFA to tumor that was many-fold higher compared with MTPA. This difference is likely predicated upon the tumor specificity of

MTPA that results in diminished hepatocyte injury. Consequently, there was a diminished release of circulating oncogenic cytokines and an absence in the accelerated growth of remote tumors. These considerations highlight the utility of a tumor-specific ablation modality that minimizes off-target thermal injury.

Ablation is an effective treatment for patients with early-stage HCC; however, the vast majority of patients present with advanced-stage disease, a diagnosis with few treatment options. Recently, though, immunotherapy through the systemic administration of immune checkpoint inhibitors has demonstrated efficacy in advanced HCC. There is a strong rationale for immunotherapy in HCC. Patients with HCC tumors expressing high levels of PD-L1 have a significantly poorer prognosis than patients with lower expression, and tumor expression of PD-L1 is an independent predictor for postoperative recurrence in patients with HCC[34]. Phase I/II trials of anti-PD-1 and anti-CTLA-4 checkpoint inhibitors for HCC have demonstrated durable responses, albeit in a minority of patients[35]. For example, the recently conducted CheckMate 040 trial, a phase I/II study of an anti-PD-1 checkpoint inhibitor for patients with advanced HCC, demonstrated an objective response rate of only 20% in the dose-expansion cohort[36]. A major question, therefore, is how to broaden treatment responses to a greater proportion of HCC patients by overcoming the barriers to immunotherapy imposed by an immunosuppressive tumor microenvironment.

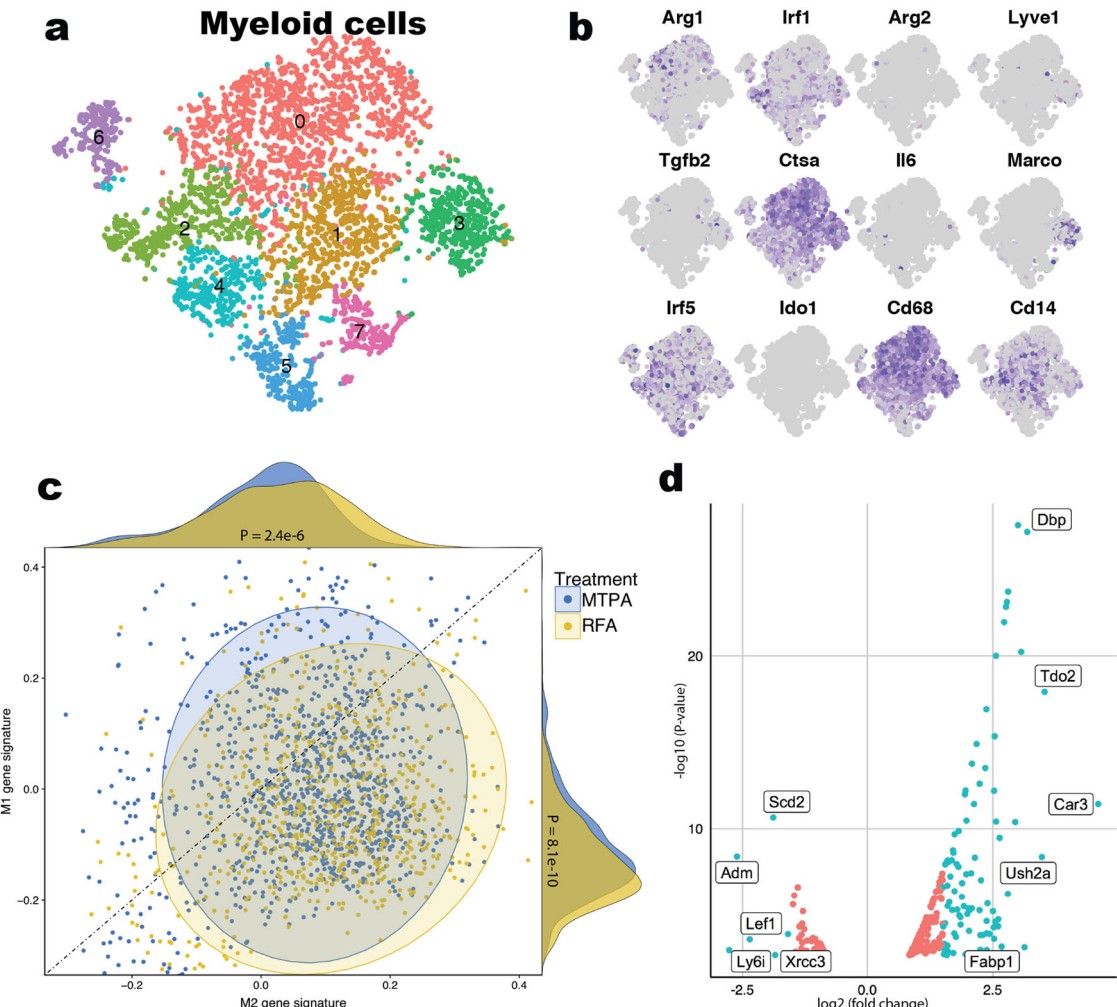

**Fig. 4 MTPA modulates the tumor immune microenvironment towards a "hot" phenotype.** scRNAseq of the intratumoral myeloid compartment was performed following sublethal hyperthermia treatment of syngeneic rat HCC tumors. Cluster characterization by marker genes (**a**, **b**) and gene set analysis (**c**) demonstrate a significant shift in myeloid cells from M2 towards an M1 phenotype with hyperthermia relative to control. In addition, bulk RNAseq following MTPA of orthotopic liver tumors was notable for upregulation of numerous interferon-stimulated genes (**d**) relative to untreated tumor.

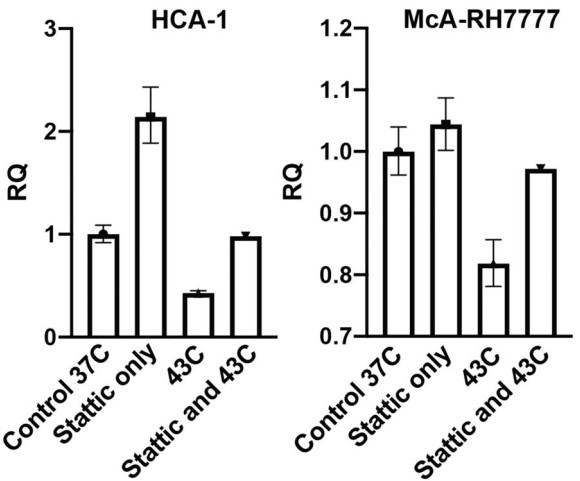

**Fig. 5 Hyperthermia decreases β-catenin expression through STAT3 signaling.** In both mouse (HCA-1) and rat (McA-RH7777) HCC cell lines, hyperthermia to 43 C decreased β-catenin expression relative to control. This effect was abrogated by the use of the STAT3 inhibitor Stattic. Each bar represents a single biological sample, and the error bars reflect variation among technical replicates.

**Table 1 Gene ontology for biological processes associated with RFA-mediated thermal injury to normal liver tissue.**

| Term | Fold enrichment | p value (Benjamini) |
|---|---|---|
| Response to heat | 20.70597166 | 3.53E-04 |
| Response to gamma radiation | 30.09439359 | 3.50E-04 |
| Response to glucocorticoid | 13.87811634 | 2.80E-04 |
| Response to estradiol | 10.33091123 | 3.19E-04 |
| Cellular response to interleukin-1 | 16.48026316 | 4.51E-04 |
| Neutrophil chemotaxis | 23.86796733 | 4.82E-04 |
| Cellular response to lipopolysaccharide | 11.05263158 | 6.34E-04 |
| Response to lipopolysaccharide | 7.416118421 | 0.001994904 |
| Response to glucose | 13.05983118 | 0.006345403 |
| Positive regulation of angiogenesis | 11.83198381 | 0.009182903 |
| Positive regulation of neutrophil chemotaxis | 36.91578947 | 0.009419599 |
| Response to hydrogen peroxide | 14.78997976 | 0.018569414 |
| Response to tumor necrosis factor | 26.36842105 | 0.02214066 |

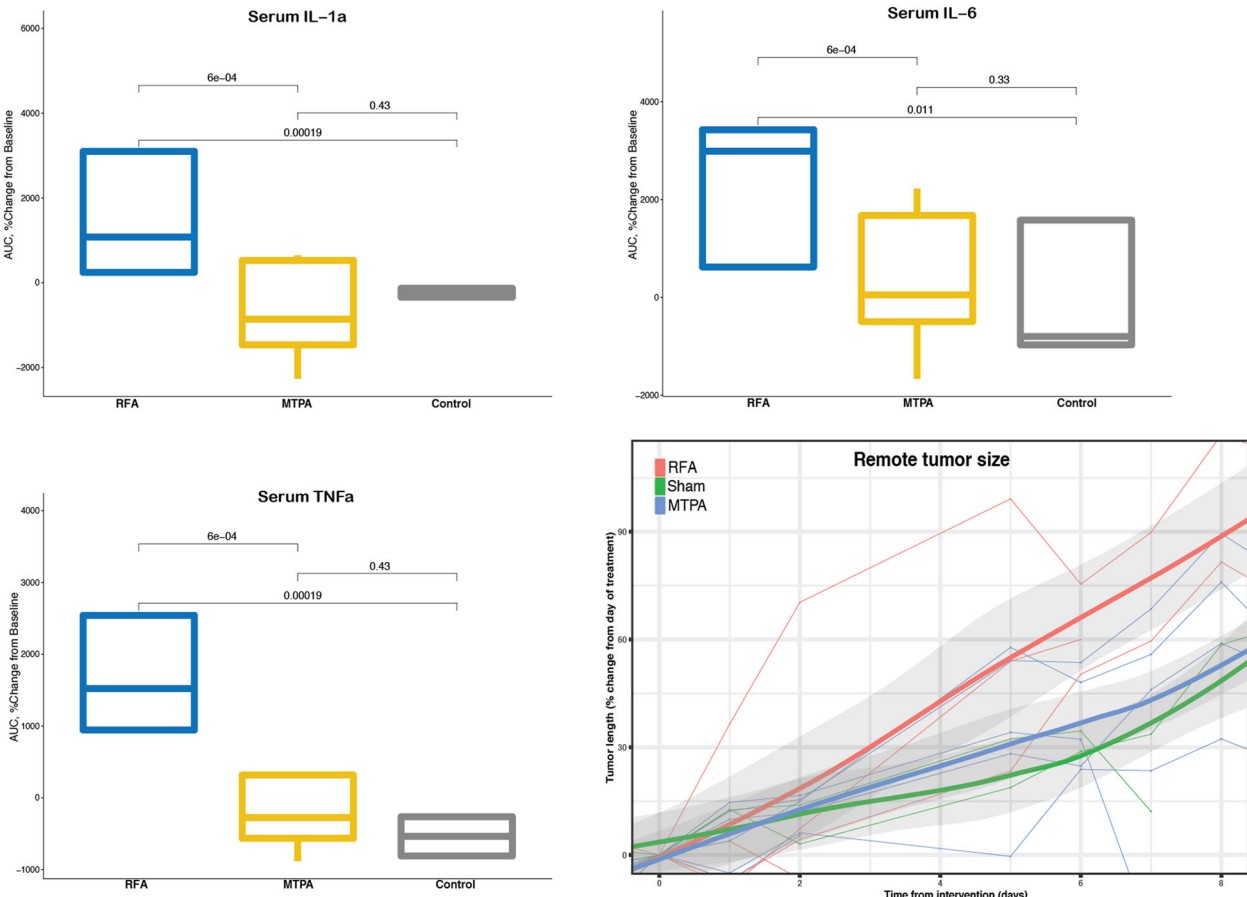

**Fig. 6 MTPA decreases systemic oncogenesis relative to RFA.** Serum cytokine profiling demonstrated that RFA elaborates significantly more systemic oncogenic cytokines relative to MTPA or sham surgery. This resulted in accelerated remote tumor growth compared with MTPA or sham. $N = 4$ animals per group with four independent blood draws per animal.

Hyperthermia is well known to counteract an immunosuppressive microenvironment. In distinction to surgery, hyperthermia causes cell stress in situ; this exposes previously shielded tumor neoantigens to the adaptive immune system. Moreover, following thermal ablation, numerous intracellular components that serve as costimulatory molecules for immune activation are released, resulting in increased number and function of intratumoral antigen-presenting cells and cytotoxic T cells[19,37,38]. Clinical trials have demonstrated that RFA can result in the expansion of tumor-specific T-cell clones[39]. Furthermore, in concordance with the current study, RFA has been shown to modulate the intratumoral myeloid population; while the frequency of myeloid-derived suppressor cells decreased in some patients, it was increased in others, and this increase was associated with worse clinical outcomes[40]. Thermal ablation has also been combined with systemic checkpoint therapy for patients with HCC. In a trial of 39 patients with intermediate and advanced stage HCC, patients underwent RFA or trans-arterial therapies in combination with tremelimumab (anti-CTLA)[41]. The combination was found to be safe, with immune profiling demonstrating a significant increase in infiltrating T cells for patients who exhibited clinical response. Numerous additional studies are currently underway to further evaluate the potential advantages of boosting systemic immunotherapy with local tumor therapies in HCC[19].

It is clear that although ablation may have favorable immunologic benefits that can augment tumor immunity, there are potential harmful immunosuppressive and oncogenic sequalae as well. The ultimate modality would be one that favors the former while minimizing the latter. We hypothesized that tumor specificity was a key component in striking this ideal balance. We found that MTPA exhibited several advantageous modulations of the tumor immune microenvironment relative to RFA. Furthermore, we identified one potential mechanism for this improvement, namely, the downregulation of β-catenin. The β-catenin pathway was recently found to be the first tumor-intrinsic oncogenic pathway to drive tumor immune escape[42]. Through the induction of the transcriptional repressor ATF3, active β-catenin downregulates CCL4 expression, resulting in failed recruitment of Batf3+ dendritic cells. β-catenin expression is a key component in the molecular classification of HCC; activating mutations of its gene *CTNNB1* define a clear subset of HCC patients and typically associated with hepatitis C or alcohol-related liver disease. Relevant to this study, tumors with driver mutations in the Wnt-β-catenin pathway are remarkable for the immune-excluded phenotype of their tumor immune microenvironment.

As shown in the current study and supported by others, not only does hyperthermia affect β-catenin levels, but it does so in a modality-dependent way[43]. That is, it is not the hyperthermia itself but rather the manner in which hyperthermia is generated that dictates β-catenin expression. Hyperthermia generated by RFA has previously been shown to increase β-catenin and in so doing increase the metastatic potential[44]. Conversely, we found that non-RF-mediated methods of hyperthermia such as in spatial profiling studies following MTPA of orthotopic tumors can suppress β-catenin levels. The interplay between hyperthermia and the Wnt/β-catenin pathway may be mediated by interferon

signaling. MTPA elaborates a type I interferon response, and interferons suppress β-catenin signaling in HCC[25]. Thus, a proposed mechanism is that the type I interferon response following MTPA suppresses β-catenin signaling and modulates the tumor immune microenvironment to promote M1 macrophage polarization and CD8+ T-cell infiltration.

Importantly, however, we identified multiple immunosuppressive changes to the tumor immune landscape regardless of the method of hyperthermia. Both MTPA and RFA increased intratumoral levels of immunosuppressive cytokines such as TGFβ as well as increased cellular expression of immune checkpoints. As seen with systemic immunotherapies, these changes likely reflect immune escape mechanisms and define resistance patterns; they also highlight one of the reasons that the "abscopal" response, wherein a locoregional therapy affects systemic tumor immunity, is vanishingly rare. Nonetheless, as many of these upregulated immune suppressors are targetable by the contemporary arsenal of immunotherapeutics, these data can help guide the selection of rational combinations of ablation tools with systemic or intratumoral therapies to maximize tumor immunity.

We acknowledge important limitations to the animal model used in this study. Although the animal model was an orthotopic, syngeneic model of HCC, tumors were implanted in animals with healthy livers rather than cirrhotic livers as is seen in humans. Although the advantage of this model is predictable size, location, and development of orthotopic tumors, the absence of underlying cirrhosis is a substantial difference from the clinical setting of HCC. The immunologic profile of the cirrhotic liver is likely significantly different compared with a healthy liver.

A key feature of the MTPA technology is its clinical translatability, though it is important to note that the current 'on-the-shelf' solutions will have to be optimized for MTPA. We have previously shown that not only does ICG localize with high TBRs to liver tumors but that this localization can be readily measured clinically at the point of care with off-the-shelf equipment[21,22,45]. The fluorescence signal detected from the tumor confirms that the target lesion has been accessed, addressing an important limitation of conventional ablation technologies. Furthermore, photothermal heat generation can readily be achieved by illuminating the target lesion in a minimally invasive manner using clinically available percutaneous NIR fibered laser catheters. Intriguingly, the localization of ICG to tumors is not unique to the liver; others have shown similar high TBRs for pulmonary tumors as well[46]. This raises the possibility for additional clinical applications beyond HCC.

## Methods

**Phantom experiments**. Inclusions of 0, 0.1, 0.2, and 0.4 mg ICG per mL gelatin solution were embedded in gelatin phantoms and ex vivo porcine liver. Each inclusion was irradiated via a fiberoptic catheter at a wavelength of 808-nm and laser powers of 0.5 W, 1 W, and 2 W. Temperature measurements were recorded using a thermocouple temperature probe (Fluoroptic, Lumasense, Santa Clara, CA).

**Syngeneic, orthotopic, immunocompetent HCC rat model**. All animal experiments were approved by The University of Texas, MD Anderson Cancer Center's Institution Animal Care and Use Committee. Animals were acquired from a Buffalo rat colony maintained at The University of Texas, MD Anderson Cancer Center. Unless otherwise indicated as a replicate measurement, data were taken from distinct samples. An HCC model was generated by orthotopic liver and/or subcutaneous implantation of McArdle RH7777 (ATCC) hepatoma cells in Buffalo rats. Orthotopic implantations will be performed by subcapsular hepatic implantation of $1 \times 10^7$ McArdle RH7777 cells via minilaparotomy. Tumor growth was monitored twice weekly with ultrasound for 1–2 weeks until the tumors were ~1cc in volume. This is a syngeneic HCC animal model in an immunocompetent rat that exhibits spontaneous hematogenous metastases.

**Animal surgeries**. All procedures were conducted under anesthesia with isoflurane (1–4%) and oxygen (1–2 L) using a rodent anesthesia machine by tank induction followed by nose cone maintenance. Pedal withdrawal reflex was used to evaluate the depth of anesthesia for gas anesthesia before beginning any invasive procedure. Animals were placed on water circulating heating pads to maintain body temperature during surgeries, and the level anesthesia was continuously monitored throughout the surgeries.

**MTPA system**. The tumoral specificity of MTPA is predicated upon the exquisite tumoral specificity of ICG for HCC. We have previously studied this specificity in both human and rodent models[21,22,45]. In the Buffalo rat model described above, animals were administered ICG 0.5 mg/kg via tail vein injection. We first performed optimization experiments to identify the post injection time point with the greatest TBRs. Imaging of NIR fluorescence within the tumors was performed using an epifluorescence imaging system previously designed by our group[21,22,45]. We identified that 96 hours following tail vein injection of ICG was the optimal time point for ICG TBRs in orthotopic tumors (Fig. 1). Therefore, this was the time point we have selected for all of the experiments described subsequently. After ICG localization, tumor-specific hyperthermia was then generated by a custom-designed system that was analogous to previously designed devices[21,22,47,48]. A 785 nm, 485 mW NIR laser (Edmund Optics) provided illumination and was coupled to a 2 mm collimating lens. Continuous thermometry will be performed using an optical sensor (Fluoroptic, Lumasense, Santa Clara, CA) and an infrared camera (FLIR, Boston, MA). MTPA of orthotopic liver tumors was performed by direct visualization following minilaparotomy and elaboration of the liver onto the skin surface. The MTPA temperature setpoint was chosen based on the experimental goal. For experiments designed to test the ability of MTPA to achieve local tumor control, the target thermal dose was 70 °C for 5 minutes. For experiments designed to evaluate the immunologic ramifications, the target thermal dose was 42–45 °C for 5 minutes.

RFA and sham surgeries were performed by creating a minilaparotomy and elaborating the liver to the surface, identical to the surgical technique for MTPA. For RFA treatment, we placed an RF needle (Radionics) in contact with the tumor, mimicking the conventional clinical approach. RFA was performed using a conventional monopolar radiofrequency pulse, which was applied using a 500-kHz generator. RF current was applied for 5 minutes with the generator output titrated to maintain a tip temperature of ~70 °C. Using these specific settings, the authors were able to reliably generate a 6–7 mm area of coagulation necrosis, which left a substantial rim of unablated tissue within the tumor. In both MTPA and RFA treatment arms, partial ablation of the tumor was performed so that viable residual tumor tissue adjacent to the ablation zone could be analyzed.

**Single-cell suspension creation and flow cytometry analysis**. The tumors were dissected and cut into small pieces with a scalpel. The tissue fragments were subject to enzymatic digestion in three volumes of PBS containing DNAse I 1 mg/mL (Roche cat# 11284932001), Collagenase D 1 mg/mL (Roche cat# 11088882001), and Dispase II 2.4 U/mL (Roche cat# 4942078001). After incubation at 37 °C for 30 min with gentle stirring, the lysate was washed with PBS+2% FBS and strained through a 40 μM nylon mesh. The red blood cells were then lysed with ammonium chloride buffer (Miltenyi cat #130-094-183). The cell suspension was washed and resuspended in PBS+2%FBS before determining the total number of cells obtained as well as their viability using Trypan blue.

The leukocytes (i.e., CD45+ cells) were then purified using CD45 MicroBeads (Miltenyi cat#130-109-682) following the manufacturer's indications. The purified cells were resuspended in ice-cold PBS+2% FBS, and the total number and viability were determined with an automatic cell counter (Countess II, Thermo Fisher).

Flow cytometry analysis was performed in the following manner. Expression of cell surface antigens was evaluated using the following antibodies: CD45-PE/Dazzle594 (BioLegend cat# 202223), CD3-VioBlue (Miltenyi cat# 130-102-677), CD8-FITC (BD Pharmingen cat# 554856), and CD4-PE/Cy7 (BD Pharmingen cat# 561578). In brief, non-specific binding was first blocked by treating the samples with an anti-CD32 antibody (BD Pharmingen, cat# 550270) at 1:200 at 4 C for 5 min. The antibody dilutions were prepared in PBS+2% FBS-containing Fixable Viability Dye eFluor™ 780 (eBioscience cat # 65-0865-14) at 1:1,000. The cells were incubated with the antibodies at the dilutions recommended by the makers during 30 minutes at 4 C in the dark. The cells were washed twice, resuspended in PBS+2%FBS, and analyzed by flow cytometry with a Gallios 561 flow cytometer (Beckman Coulter, CA).

**Serum cytokine analysis**. Blood samples were collected from rats treated with RFA, MTPA, and sham surgeries at multiple timepoints (pre-procedure as well as 3-, 24-, and 48 hours post procedure). Samples were kept on ice and then centrifuged at 4500 rpm for 3 minutes at 4 °C. The supernatant was collected and stored at −80 °C until analysis was performed using the Luminex rat multiplex cytokine panel (Luminex, Austin, TX) according to the manufacturer's guidelines. To quantify the time course of cytokine changes following each intervention, an area under the curve analysis was performed[49]. This approach allows for the evaluation of dynamic changes from baseline values over time. Statistically significant differences in area under the curve values between each experimental group was determined using the Wilcoxon test.

**In vitro hyperthermia treatment and evaluation of gene expression.** The mouse HCC cell line HCA-1 (kindly provided by Dr. Dan Duda, Harvard Medical School, Charlestown MA) and the rat hepatoma cell line McA-RH7777 (ATCC, Manassas VA) were subject to sublethal hyperthermia in vitro and the associated changes in gene expression were evaluated with quantitative RT-PCR. In all, 80% confluent cultures were treated with or without 5 μM of the Stat3 inhibitor Stattic (MedChem Express, Monmouth Junction NJ) in 2% fetal bovine serum (FBS)-containing media and incubated at 37 °C for 4 hours. At that point, one of the non-treated control plates and one of the Stattic-treated plates were transferred to an incubator set at 43 °C for additional 45 minutes. mRNA was collected using RLT Buffer from the Qiagen RNeasy kit (Qiagen, Germantown MD). cDNA was synthesized with the Bio-Rad cDNA iScript Kit (Bio-Rad, Hercules CA). qPCR evaluation of Ctnnb1 (beta-catenin 1) was performed with Mouse Ctnnb1 TaqMan® Gene Expression Assay (Assay ID: Mm00483039_m1), Mouse Actb Taqman Gene Expression Assay (Assay ID: Mm00607939_s1, and Taqman Fast Advanced Master Mix (Applied Biosystems, Austin TX). qPCR for evaluation of Ctnnb1 (beta-catenin 1) was performed with Rat Ctnnb1 Taqman Gene Expression Assay (Assay ID: Rn00670330_m1) and Eukayotic 18 S rRNA Taqman Gene Expression Assay (Assay ID: HS99999901_s1), and Taqman Fast Advanced Master Mix (Applied Biosystems, Austin TX).

**Nanostring DSP.** Protein expression within prescribed tissue territories following MTPA and RFA was performed using the commercially available NanoString GeoMX Digital Spatial Profiler (DSP) technology (Table S1). Formalin-fixed, paraffin-embedded slides of liver tumors treated with MTPA, RFA, or sham surgery were first stained with fluorescently labeled antibodies (anti-CD3, anti-CD8, anti-pan-cytokeratin, anti-DNA) to delineate the locations of relevant cell populations within the tissue. Regions of interest (ROIs) were then manually drawn over the fluorescence microscopy images of the tissue sections within specific territories of interest: untreated tumor, liver tumor boundary, ablated tumor, ablation margin, and normal liver. Within these territories, a panel of 48 anti-rat antibodies was used to perform quantitative, single-analyte precision measurements of immunologically relevant proteins. Normalization of expression levels was performed both by ROI area as well as by expression of housekeeper genes (S6, GADPH). Differences in protein expression across tissue territories were performed using the Wilcoxon rank-sum test.

**Tumor growth measurements.** For subcutaneous tumors, the tumor size was measured using a digital caliper and tumor volume was calculated using the formula $V = 0.5*(height)*(width^2)$.

**Bulk and single-cell RNA-sequencing analysis.** All genomic analysis was performed within the R environment (R Foundation). Bulk RNA sequencing (RNA-seq) was performed on liver and tumor tissue in the following manner. Tissue was harvested following MTPA, RFA, or sham surgery and was stored at −80 °C until analysis was performed using the Illumina platform. Read alignment and transcript quantitation was performed by the kallisto package. Differentially expressed genes between the treatment groups were calculated using the DeSeq2 package. Functional analysis of these differentially expressed genes was performed using the DAVID toolkit[27,28].

For detailed characterization of the tumor immune microenvironment, single-cell RNA sequencing (scRNAseq) was performed in the following manner. Orthotopic liver tumors were harvested following MTPA, RFA, or sham surgery. CD45+ cells were isolated into single-cell suspensions as previously described. The scRNAseq libraries were then prepared by the 10X Genomics Chromium single Cell Immune Profiling Solution based on the manufacturer's recommendations using 10,000 purified leukocytes per sample. The cDNA of single cell transcriptomes were then sequencing using the Illumina platform (NextSeq 500). The data were then analyzed using the Seurat package[50]. We first constructed an atlas of CD45+ cells that underwent scRNAseq. After identifying the major immune cell types, clusters were annotated based upon established transcriptional profiles[51]. Next, we subselected the clusters representing the myeloid populations (macrophages and monocytes) and characterized their phenotypic state using previously published gene sets defining classical "M1" and "M2" phenotypes[52].

**Statistics and reproducibility.** All genomic analysis was performed within the R environment (R Foundation, version 4.0.2). Sample sizes and statistical methods were as described for the individual experiments above. All cell lines were negative for mycoplasma or other contaminants by MAP testing (PCR method).

**Reporting summary.** Further information on research design is available in the Nature Research Reporting Summary linked to this article.

## Data availability

Sequence data that support the findings of this study have been deposited in GenBank with the primary accession code PRJNA668800. All other data generated during and/or analysed during the current study are available from the corresponding author on reasonable request.

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

## Acknowledgements

This work was supported by funds from the Society for Interventional Radiology Foundation, the Radiological Society of North America Research and Education Foundation, and NIH/NIBIB (R21 EB026089-01A1). We acknowledge the use of The University of Texas, MD Anderson Cancer Center's Flow Cytometry and Cellular Imaging Core Facility (FCCICF), which is partially funded by the NCI Cancer Center Support Grant (P30CA16672). A.R. was supported by institutional startup funds from the University of Michigan, NCI grant R37CA214955 and a Research Scholar Grant from the American Cancer Society (RSG-16-005-01-CCE).

## Author contributions

R.A.S. and N.M. conceived the idea, designed experiments, analyzed data, and wrote the manuscript. R.A.S., N.M., M.W., K.D., C.D., A.M., R.A., R.B., K.S., and A.R. analyzed data and performed experiments. R.B., K.S., and A.R. assisted with data processing and data presentation. A.K. provided conceptual advice and technical support. R.A.S. supervised the study. All authors discussed the results and reviewed on the manuscript.

## Competing interests
The authors declare no competing interests.
