## [Peer Review File · Communications Biology]

Reviewers' Comments:

Reviewer #1:

Remarks to the Author:

Munoz et al described novel thermal ablation approach to treat HCC using molecular targeted therapy (MTPA). They have shown in vitro and in vivo the specificity of this approach and more importantly the favorable immune modulation effect of MTPA compared to RFA. This is well written report and a novel approach, however the mechanism by which MTPA was speculated to modulate the immune microenvironment needs to be further characterized. Specifically, the authors hypothesis is that MTPA can increase type I interferon which can downregulate beta-catenin that is known to drive immune resistance. The authors need to provide more supportive data to prove this hypothesis. they could consider knocking down type I interferon and test whether the MTPA effect on beta-catenin remains valid.

in addition, the landscape of HCC treatment has shifted to the combination of immunotherapy and anti-angiogenesis with the recent approval of atezolizumab and bevacizumab. the authors need should consider testing the effect of MTPA on angiogenesis at the molecular and genomics level in order to position this therapy within the current landscape.

Reviewer #2:

Remarks to the Author:

The authors have submitted their manuscript, COMMSBIO-20-1461-T, entitled "Molecularly targeted photothermal ablation for local control and immunologic modulation of hepatocellular carcinoma". Overall, this is an excellently executed study on MTPA as a potential thermal ablation modality and its effects on both downstream pro-immune and pro-oncogenic effects following ablation. Comments are outlined below:

Strengths:

1. This an excellently executed paper that systematically demonstrates the specificity of MTPA as a locally ablative tumor treatment, and it's downstream influence both towards a pro-immune tumor microenvironment, and while simultaneously reducing pro-oncogenic effects seen following thermal ablation.
2. Very well-written manuscript.

Weaknesses:

3. [Minor] The authors highlight a concern regarding inability to 'localize' the target tumor with conventional methods that is likely over-stated in the Introduction and in the Discussion. Numerous studies and recent advances in image navigation technology make confirmation of intratumoral needle location less clinically-relevant.

Section-specific comments:

4. Introduction: Well-written. Clearly articulated need for more specificity beyond that which is offered with conventional tumor thermal ablation (both with regard to immediate tumor treatment and from secondary effects of thermal heating on periablational normal tissue parenchyma).
5. Results: Logical progression of studies.
6. Discussion: The authors discuss salient results in the context of current thermal ablative therapies. While long, potentially likely required given diverse readership. (minor)
7. Discussion: The authors should specifically discuss limitations of the study. For example, use of an orthotopic model such as RF-7777 in normal liver are not necessarily translatable to HCC in cirrhotic liver (in particular as the animal model in Buffalo rats may be more immunoresponsive, and cirrhotic liver has a different baseline immune environment than normal liver). Similarly, clinical translatability of current platforms is an additional limitation. (minor)

Referee expertise:

Referee #1: Cancer Immunotherapy, GI tumors

Referee #2: Thermal ablation therapies, cancer treatments

Reviewers' comments:Reviewer #1 (Remarks to the Author):

Munoz et al described novel thermal ablation approach to treat HCC using molecular targeted therapy (MTPA). They have shown in vitro and in vivo the specificity of this approach and more importantly the favorable immune modulation effect of MTPA compared to RFA. This is well written report and a novel approach, however the mechanism by which MTPA was speculated to modulate the immune microenvironment needs to be further characterized. Specifically, the authors hypothesis is that MTPA can increase type I interferon which can downregulate beta-catenin that is known to drive immune resistance. The authors need to provide more supportive data to prove this hypothesis. they could consider knocking down type I interferon and test whether the MTPA effect on beta-catenin remains valid. In addition, the landscape of HCC treatment has shifted to the combination of immunotherapy and anti-angiogenesis with the recent approval of atezolizumab and bevacizumab. the authors need should consider testing the effect of MTPA on angiogenesis at the molecular and genomics level in order to position this therapy within the current landscape.

We thank the Reviewer for their thoughtful and constructive comments. In particular, we find the recommendation to further define the molecular mechanisms behind MTPA's immunomodulatory effects very valuable. To address this limitation in the original manuscript, we have conducted additional experiments that are included in the revision. Specifically, we investigated the interplay between MTPA's hyperthermia effect on the interferon - STAT3 pathway and beta-catenin expression. As we demonstrate in the new figures, hyperthermia resulted in a decrease in beta catenin expression, and this effect was abrogated with a STAT inhibitor, highlighting the interplay between hyperthermia and this signaling pathway. This effect was reproducible in two HCC cell lines from two different species.

We also agree that the landscape of HCC systemic therapies has evolved rapidly in the past 12 months alone, and that tyrosine kinase inhibitors and other VEGF inhibitors are central to HCC therapy. The influence of MTPA on the tumor microvasculature is indeed a compelling question; however, we feel that this question is outside of the scope of this manuscript's primary focus which is on the immunologic ramifications. We look forward to exploring the microvascular changes following MTPA in future experiments, and we would be happy to include this concept in the Discussion section if so desired.

Reviewer #2 (Remarks to the Author):

The authors have submitted their manuscript, COMMSBIO-20-1461-T, entitled "Molecularly targeted photothermal ablation for local control and immunologic modulation of hepatocellular carcinoma". Overall, this is an excellently executed study on MTPA as a potential thermal ablation modality and its effects on both downstream pro-immune and pro-oncogenic effects following ablation. Comments are outlined below:

Strengths:

1. This an excellently executed paper that systematically demonstrates the specificity of MTPA as a locally ablative tumor treatment, and it's downstream influence both towards a pro-immune tumor microenvironment, and while simultaneously reducing pro-oncogenic effects seen

following thermal ablation.

2. Very well-written manuscript.

We thank the Reviewer for their supportive comments.

Weaknesses:

3. [Minor] The authors highlight a concern regarding inability to 'localize' the target tumor with conventional methods that is likely over-stated in the Introduction and in the Discussion. Numerous studies and recent advances in image navigation technology make confirmation of intratumoral needle location less clinically-relevant.

We thank the Reviewer for their very helpful comment. We agree that new image guidance tools have substantially improved techniques for minimally invasive hepatic interventions. We have revised the manuscript to de-emphasize these challenges.

Section-specific comments:

4. Introduction: Well-written. Clearly articulated need for more specificity beyond that which is offered with conventional tumor thermal ablation (both with regard to immediate tumor treatment and from secondary effects of thermal heating on periablational normal tissue parenchyma).

We thank the Reviewer for their supportive comments.

5. Results: Logical progression of studies.

We thank the Reviewer for their supportive comments.

6. Discussion: The authors discuss salient results in the context of current thermal ablative therapies. While long, potentially likely required given diverse readership. (minor)

We thank the Reviewer for their supportive comments. We have kept the Discussion at the same length in the revision as it remains within the suggested word limits of the Journal.

7. Discussion: The authors should specifically discuss limitations of the study. For example, use of an orthotopic model such as RF-7777 in normal liver are not necessarily translatable to HCC in cirrhotic liver (in particular as the animal model in Buffalo rats may be more immunoresponsive, and cirrhotic liver has a different baseline immune environment than normal liver). Similarly, clinical translatability of current platforms is an additional limitation. (minor)

We thank the Reviewer for their helpful comments. We have added these appropriately identified limitations to the Discussion section.

Reviewers' Comments:

Reviewer #1:

Remarks to the Author:

the authors have kindly addressed my comments